# High-Glucose-Induced Metabolic and Redox Alterations Are Distinctly Modulated by Various Antidiabetic Agents and Interventions Against FABP5/7, MITF and ANGPTL4 in Melanoma A375 Cells

**DOI:** 10.3390/ijms26031014

**Published:** 2025-01-24

**Authors:** Nami Nishikiori, Hiroshi Ohguro, Megumi Watanabe, Megumi Higashide, Toshifumi Ogawa, Masato Furuhashi, Tatsuya Sato

**Affiliations:** 1Departments of Ophthalmology, Sapporo Medical University, S1W17, Chuo-ku, Sapporo 060-8556, Japan; nami076@yahoo.co.jp (N.N.); watanabe@sapmed.ac.jp (M.W.); megumi.h@sapmed.ac.jp (M.H.); 2Departments of Cardiovascular, Renal and Metabolic Medicine, Sapporo Medical University, S1W17, Chuo-ku, Sapporo 060-8556, Japan; a08m024@yahoo.co.jp (T.O.); furuhasi@sapmed.ac.jp (M.F.); 3Departments of Cellular Physiology and Signal Transduction, Sapporo Medical University, S1W17, Chuo-ku, Sapporo 060-8556, Japan

**Keywords:** malignant melanoma, metformin, imeglimin, extracellular flux analyzer, ROS, MITF, FABP

## Abstract

Hyperglycemia-induced effects on cellular metabolic properties and reactive oxygen species (ROS) generation play pivotal roles in the pathogenesis of malignant melanoma (MM). This study assessed how metabolic states, ROS production, and related gene expression are modulated by antidiabetic agents. The anti-diabetic agents metformin (Met) and imeglimin (Ime), inhibitors of fatty acid-binding proteins 5/7 (MF6) and microphthalmia-associated transcription factor (MITF) (ML329), and siRNA-mediated knockdown of angiopoietin-like protein 4 (ANGPTL4), which affect mitochondrial respiration, ROS production, and related gene expression, were tested in A375 (MM cell line) cells cultured in low (5.5 mM) and high glucose (50 mM) conditions. Cellular metabolic functions were significantly and differently modulated by Met, Ime, MF6, or ML329 and knockdown of ANGPTL4. High glucose significantly enhanced ROS production, which was alleviated by Ime but not by Met. Both MF6 and ML329 reduced ROS levels under both low and high glucose conditions. Knockdown of ANGPTL4 enhanced the change in glucose-dependent ROS production. Gene expression related to mitochondrial respiration and the pathogenesis of MM was significantly modulated by different glucose conditions, antidiabetic agents, MF6, and ML329. These findings suggest that glucose-dependent changes in cellular metabolism and redox status are differently modulated by antidiabetic agents, inhibition of fatty acid-binding proteins or MITF, and ANGPTL4 knockdown in A375 cells.

## 1. Introduction

Melanocytes primarily function to protect the skin from UV-radiation-induced damage by their synthesis and secretion of melanin. Malignant melanoma (MM), a type of skin cancer that originates from melanocytes, can arise from not only the skin but also various other tissues and organs, including the gastrointestinal tract, genitalia, sinuses and ocular choroid [1,2,3]. As for the pathogenesis of MM at the genetic level, mutations in BRAF and NRAS genes, genes that activate the RAS/mitogen-activated protein kinase (MAPK) pathway, are detected in approximately 40–50% and 15–20% of human MM cases, respectively [4]. Therefore, MAPK signaling, which contributes to the initiation of MM and progression to metastasis, is hyperactivated in most cases of human cutaneous MM [5], and this evidence has been applied to the development of new MM therapies using various targeted inhibitors for BRAFV600E and MAPK kinase (MEK), as well as immune-based strategies, resulting in remarkable improvements in the survival and quality of life in the past 10 years [4,6,7,8].

Recently, metabolic dysfunction conditions such as diabetes, obesity and postprandial hyperglycemia have been shown to be recognized as distinct risk factors for various cancers, including colorectal and breast cancers in addition to MM [9,10,11]. One of the possible underlying mechanisms of the hyperglycemia-induced effects on cancerous cells is the “Warburg effect”, which is characterized by activated glycolytic flux to produce lactate even in the presence of sufficient oxygen during tumor progression [12]. In cases of MM, it has been shown that elevated glucose uptake induces the hyper-activated MAPK pathway, thereby suppressing oxidative phosphorylation (OXPHOS) and enhancing glycolysis [13,14]. Although it has not been shown how such uncoupled enhancement of glycolysis plays roles in the pathogenesis of MM, a previous study showed that lowering high glucose concentrations to physiological levels to mimic restriction of glucose availability stimulated the production of reactive oxygen species (ROS), supporting the notion that larger glycemic variability can increased the risk of MM [15]. In addition, an increase in the generation of ROS in response to a restriction of glucose utilization may be caused by a microphthalmia-associated transcription factor (MITF)-related mechanism; MITF is known to be involved in the regulation of cell survival, proliferation and invasion and the drug resistance of MM [15]. However, in contrast, it has been reported that levels of ROS are increased in response to high glucose levels in non-cancerous cells such as 3T3-L1 preadipocytes [16], retinal pigment epithelium cells [17], Schwann cells [18] and H9c2 cardiomyoblasts [19], as well as cancerous cells such as pancreatic carcinoma cells [20], breast cancer cells [21] and colorectal cancer stem cells [22]. These collective findings suggest that the relationship between the generation of ROS and glucose levels is not consistent, indicating that additional factors such as lipid regulators and hypoxia-induced pathways are also involved in the glucose-dependent pathogenesis of MM. In fact, fatty acid-binding protein 4 (FABP4) was shown to be involved in the high-fat-diet-induced generation of ROS during tumorigenesis of a mammary tumor [23], whereas FABP5 was suggested to be involved in the metastasis of MM [24]. In addition, angiopoietin-like protein 4 (ANGPTL4), which is upregulated by metabolic demand, has been shown to be involved in the pathogenesis of angiogenesis of the placenta [25] and diabetes-related complications [26,27] and also to promote the metastasis of MM [28]. However, as of the time of writing, the contribution of FABPs and/or ANGPTL4 to glucose-dependent biological changes in the pathogenesis of MM have not been studied in detail.

Metformin (Met) is a conventional antidiabetic agent that has been widely used for treatment of patients with type 2 diabetes and prediabetic conditions due to its good tolerability and low cost. An adenosine 5′-monophosphate (AMP)-activated protein kinase (AMPK)-dependent mechanism has been shown to be primarily involved in the reduction of hyperglycemic condition by Met [29,30]. Met also has unique AMPK-independent actions [31], including additional and unexpected biological functions to inhibit inflammation, the generation of ROS, the senescence of endothelial cells, programmed cell death and neovascularization, in addition to its hypoglycemic action [32,33]. In fact, recent studies have suggested that the unique pharmacological effects of Met may lower the risk of cancer in patients with diabetes, thereby prolonging the survival rate of cancer patients [34,35,36]. Therefore, Met may have potential as an adjuvant for the treatment of some cancers, including MM. In addition to Met, imeglimin (Ime), the first approved drug in a new Tetrahydrotriazine class of oral antidiabetic agents, is expected to solve various unmet medical needs for patients with diabetes [37,38]. In fact, recent clinical trials in Japanese and Caucasian patients with type 2 diabetes have shown that Ime has a favorable and durable antihyperglycemic action with safety, tolerability and a lack of severe hypoglycemia [39,40,41]. Interestingly, it has also been shown that Ime exerted pharmacological effects similar to those of Met on mitochondrial respiration, AMPK activity and gene expression in cultured hepatocytes [42] and that Ime effectively reduced the production of ROS in diabetic models [43,44], suggesting that Ime has anticancer effects similar to those of Met.

Therefore, to elucidate the unidentified effects of the antidiabetic regents, FABP5, ANGPTL4 and/or MITF on diabetic states of MM, high-glucose-induced metabolic and redox alterations were studied using melanoma A375 cells. That is, the effects of treatment with Ime on the cellular biological properties, including generation of ROS and metabolic functions, were compared with the effects of treatment with Met in high glucose conditions. In addition, the contribution of MITF, FABPs and ANGPTL4 to the high-glucose-induced biological alterations of A375 cells was investigated.

## 2. Results

The purpose of the present study was to elucidate the unidentified effects of the antidiabetic agents immeglimin (Ime) and metformin (Met), FABP family proteins, ANGPTL4 and MITF on biological aspects including the diabetic states of MM, as assessed by evaluating the high-glucose-induced metabolic and redox alterations in melanoma A375 cells.

### 2.1. Detection of mRNA Expression of FABP5, FABP7 and ANGPTL4 in 375 Cells

Initially, mRNA expression of *FABP* family proteins including *FABP3*, *FABP4*, *FABP5* and *FABP7* and *ANGPTL4* was evaluated in A375 cells by qPCR. As shown in Figure 1, among the *FABP* family proteins, *FABP5* and *FABP7*, but not *FABP3* and *FABP4*, were detected; *ANGPTL4* was also detected.

### 2.2. Cytotoxicity of the Antidiabetic Agents Met and Ime and Specific Inhibitors for FABP5/7 and MITF in A375 Cells Under Different Glucose Conditions

To elucidate the possible contributions of FABPs and ANGPTL4 to biological aspects of A375 cells under low glucose conditions and high glucose conditions, the effects of pharmacological inhibition of FABP5 and FABP7 using MF6, a specific inhibitor for both FABPs, and siRNA-mediated knockdown of ANGPTL4 on cellular metabolic functions were assessed by using a Seahorse bioanalyzer, and levels of ROS were measured. In addition, to elucidate the contribution of MITF, a critical transcriptional factor, in these experimental conditions, ML329, a specific inhibitor of MITF, was used. Before carrying out these analyses, we verified negligible levels of toxicity in A375 cells in the presence (1) 2 mM Met, (2) 2 mM Ime, (3) 10 μM MF6 and (4) 10 μM ML329 under low (5.5 mM) and high (50 mM) glucose conditions (Figure 2).

### 2.3. Effects of the Antidiabetic Agents Met and Ime, Specific Inhibitors for FABP5/7 and MITF or SiRNA Knockdown of ANGPTL4 on Cellular Metabolic Function in A375 Cells Under Different Glucose Conditions

To elucidate the pharmacological effects of Met, Ime, ML329 and MF6 and the effect of siRNA-mediated knockdown of ANGPTL4 on cellular metabolic functions under low and high glucose conditions, metabolic indices of mitochondrial and glycolytic functions were evaluated using an extracellular flux analyzer. In the comparison between pharmacological interventions (Figure 3), there were no statistically significant differences in the metabolic indices of mitochondrial and glycolytic functions between a low glucose condition and a high glucose condition. While both Met and Ime significantly decreased most of indices of mitochondrial function, glycolytic function was predominantly inhibited by Ime but not by Met (Figure 3). Although MF6 and ML329 did not affect mitochondrial functions, glycolytic functions tended to be decreased in response to MF6 and ML329 only under low glucose conditions (Figure 3).

In the comparison between control and ANGPTL4 siRNA (Figure 4), knockdown of ANGPTL4 induced a significant metabolic shift from glycolysis to mitochondrial respiration under a high glucose condition. Collectively, the results indicate that the mitochondrial and glycolytic functions of A375 cells were significantly and differently modulated by antidiabetic agents, an FABP inhibitor, an MITF inhibitor and knockdown of ANGPTL4**.**

### 2.4. Effects of the Antidiabetic Agents Met and Ime, Specific Inhibitors for FABP5/7 and MITF or SiRNA Knockdown of ANGPTL4 on Levels of ROS in A375 Cells Under Different Glucose Conditions

To further study the effects of hyperglycemic stimulation on oxidative stress in A375 cells, the levels of ROS were measured under low and high glucose conditions. As shown in Figure 5, the levels of ROS were significantly increased under a high glucose condition compared under a low glucose condition. These results are different from the results of a previous study using MM cell lines [15] but the same as the results of previous studies using various non-cancerous cells [16,17,18,19] and non-MM cancerous cells [20,21,22]. Such elevated levels of ROS in a high glucose condition were significantly decreased by Ime but not by Met, suggesting that Ime has a potent effect on elevated levels of ROS. MF6 and ML329 decreased the levels of ROS in both a low glucose condition and a high glucose condition. Interestingly, knockdown of ANGPTL4 enhanced the change in glucose-dependent production of ROS (Figure 5D). These findings suggest that a high glucose condition can increase production of ROS and that the increase in the production of ROS can be modulated by antidiabetic agents, FABP inhibitors, an MITF inhibitor, and knockdown of ANGPTL4.

To estimate the mechanisms underlying the effects on cellular metabolic functions and levels of ROS in A375 cells by antidiabetic reagents, pharmacological inhibition of FABPs and MITF and knockdown of ANGPTL4, mRNA expression levels of possible factors related to mitochondrial function and MM pathogenesis, including *PGC1α*, *HIF1α*, *SAMD3*, *TRIB3* and *ANGPTL4*, were evaluated by qPCR analysis. As shown in Figure 6, a high glucose condition induced significant upregulation of the mRNA expression of *PGC1α* and downregulation of the mRNA expression of *TRIB3*. Met induced significant downregulation of the mRNA expression of *TRIB3* and *ANGPTL4*, and Ime caused substantial upregulation of the mRNA expression of *PGC1α*, *HIF1α* and *SMAD3* and substantial downregulation of the mRNA expression of *TRIB3* and *ANGPTL4*. Additionally, (1) MF6 caused significant downregulation of the mRNA expression of *PGC1α*, *HIF1α* and *SMAD3* in a low glucose condition and significant upregulation of the mRNA expression of *ANGPTL4* in a high glucose condition and (2) ML329 induced significant downregulation of the mRNA expression of *PGC1α* and *TRIB3* and significant upregulation of *HIF1α* and *SMAD3* in a high glucose condition. Collectively, the results obtained in the present study showed that (1) hyperglycemic stimulation induced a significant increase in the levels of ROS in A375 cells that were beneficially reduced by Ime but not Met and (2) the effects hyperglycemic stimulation on oxidative stress were substantially modulated by FABP5 and 7, MITF and ANGPTL4.

## 3. Discussion

Recent observations suggest that increases in levels of ROS, oxidative stress, and redox imbalance are related to the genesis of MM [45,46,47,48]. In fact, epidermal melanocytes are known to be vulnerable to oxidative stress since the ROS generated during melanin biosynthesis and ultraviolet A (UVA) radiation [49] induce DNA and lipid damage, thereby inducing the generation of tumor-initiating cells due to DNA reparation/apoptosis [46,50]. Therefore, redox homeostasis regulating oxidative stress and the antioxidant response balance is a possible underlying mechanism for the initiation and following progressive processes of MM, as well as the drug resistance of MM [51,52], suggesting that the level of ROS in MM cells may be a critical biomarker of MM pathogenesis. In the present study, we found that levels of ROS were significantly increased in a high glucose condition, and the high-glucose-induced generation of ROS was significantly reduced by Ime but not by Met. In addition, pharmacological suppression of FABP5 and FABP7 by MF6 and pharmacological suppression of MITF by ML329 decreased levels of ROS in both a low glucose condition and a high glucose condition. Collectively, the results suggested that the antidiabetic reagent Ime and pharmacological suppression of FABPs and MITF may be therapeutic strategies for MM.

Met is one of the first-line antidiabetic reagents for type 2 diabetes (T2DM). Met causes hepatic gluconeogenesis to be inhibited due to its stimulatory effect on AMPK. Met-induced activation of AMPK inhibiting mitochondrial respiratory chain complex I results in a reduction in ATP production, leading to an increase in the AMP/ATP ratio [53,54]. Reduced levels of ROS are known to be caused by an AMPK-independent mechanism of Met, and it was shown that Met could reduce production of ROS to protect cells from DNA damage and mutagenesis [55]. Since it was shown that Ime also induced AMPK activity similarly to Met in hepatocytes but that two drugs differ in their effects on the mitochondrial function [42], it was rationally speculated that Ime causes a reduction of ROS levels in MM. In the present study, we found that Ime has a more potent effect than that of Met on elevated levels of ROS under high glucose conditions. In support of this, analysis of Seahorse cellular metabolic measurements showed that mitochondrial functions were also more effectively suppressed by Ime than Met.

Among FABPs, FABP5 was shown to be overexpressed in several tumor types and its expression level was shown to be associated with the growth and metastasis of several cancer types including prostate cancer, intrahepatic cholangiocarcinoma, colorectal cancer and cervical cancer [56,57,58,59]. In a recent study using a single-cell RNA-based spatial molecular imaging (RNA-SMI) technique, increased levels of FABP5 expression were detected in proliferative fields of primary and metastatic melanoma versus nevi, in addition to FABP5 expression in keratinocytes of the upper layers of the epidermis and in other cell types [60]. Furthermore, it was shown that FABP5 expression was positively correlated with progression-associated clinicopathological factors and poor prognosis in uveal melanoma (UVM), suggesting that FABP5 may be a possible oncogene and prognostic marker in patients with UVM [61]. A recent in vivo study using a zebrafish melanoma model showed that MITF regulates the antioxidant program, thereby increasing the survival of melanoma cell lines by protecting the cells from damage induced by ROS [62]. Alternatively, expression of ANGPTL4 was shown to be regulated by hypoxia in tumor cells [63], and ANGPTL4 is highly expressed in melanoma brain metastasis, suggesting that ANGPTL4 is involved in melanoma metastasis [64].

We acknowledge that the current study has several limitations. Firstly, the precise linkage mechanisms by which cellular metabolic functions and levels of ROS are modulated by antidiabetic reagents, FABPs, MITF and ANGPTL4 in diabetic states of A375 cells remain to be elucidated. Secondly, we do not know why the results for the levels of ROS under different glucose conditions were different from the results of a previous study using MM cell lines [15]. Thirdly, the effects of antidiabetic agents, FABPs, MITF and ANGPTL4 on cellular metabolism and redox status have not characterized in detail. Fourthly, as of the time of writing, our results have little relevance to further understanding the pathogenesis of or developing new therapeutic strategies for MM. Therefore, further investigation to elucidate the unidentified mechanisms influencing the effects of antidiabetic agents, FABPs, MITF and ANGPTL4 on diabetic states of MM by characterizing their down-stream signals using in vivo system such as a diabetic animal model needs to be conducted in the future.

## 4. Materials and Methods

### 4.1. Preparation of Two-Dimensionally Cultured A375 Cells

Experiments using the commercially available A375 human melanoma cell line (CRL-1619™, American Type Culture Collection, Manassas, VA, USA) were approved by the internal review board of Sapporo Medical University (# 312-3190). Briefly, A375 cells were cultured in a two-dimensional (2D) manner and maintained in 2D culture dishes of 150 mm in diameter in a low-glucose (LG: 5.5 mM glucose) DMEM medium supplemented with 10% FBS, 1% L-glutamine and 1% antibiotic-antimycotic via daily changing of the medium under standard normoxia conditions (37 °C, 5% CO_2_). To study the effects of high glucose (50 mM) and pharmacological interventions with dimethyl sulfoxide (DMSO) as a vehicle control, Met (2 mM), Ime (2mM), MF6 (as an inhibitor of FABP5 and FABP7) (10 μM), and the MITF inhibitor ML329 (10 μM), planar cultured A375 cells were incubated with a high concentration of glucose and the aforementioned agents for 24 h until analysis using the methods described below. For knockdown of ANGPTL4, 2D-cultured A375 cells were incubated with Lipofectamine 3000 (Invitrogen, Carlsbad, CA, USA) containing 30 nM of siRNA-ANGPTL4 (Merck, Rahway, NJ, USA) for 24 h according to the manufacturer’s instructions.

### 4.2. Cell Viability Assay

A commercially available kit (Cell Counting Kit-8, Dojindo, Tokyo, Japan) was used to determine the viability of A375 cells. Briefly, following the incubation of the cells with 10 μL of a reactive reagent for 2 h, measurement of the absorbance at 450 nm was carried out using a microplate reader (multimode plate reader EnSpire^®^, PerkinElmer, Waltham, MA, USA).

### 4.3. Extracellular Flux Assay

Following treatment of the planar-cultured A375 cells with various agents including Met, Ime, MF6 and ML329 or transfection of siRNA against ANGPTL4, approximately 20,000 cells per well in were plated in Seahorse XF96 cell culture microplates (#103794-100, Agilent Technologies, Santa Clara, CA, USA). After overnight incubation, the medium was replaced with 180 μL of Seahorse XF DMEM assay medium containing 5.5 mM glucose, 2.0 mM glutamine, 1.0 mM sodium pyruvate and 5.0 mM HEPES (pH 7.40), and the assay plate was further incubated in a CO_2_-free incubator at 37 °C for 45 min prior to measurements.

The oxygen consumption rate (OCR) and extracellular acidification rate (ECAR) were measured on an extracellular flux analyzer (Seahorse XFe96 Bioanalyzer, Agilent Technologies, Santa Clara, CA, USA) under a 3 mins mix and 3 mins measuring protocol at baseline and following sequential injection of oligomycin (final concentration: 2.0 μM), carbonyl cyanide p-trifluoromethoxyphenylhydrazone (FCCP, final concentration: 5.0 μM), rotenone/antimycin A mixture (final concentration: 1.0 μM) and 2-deoxyglucose (2DG, final concentration: 10 mM).

### 4.4. Measurement of Levels of Reactive Oxygen Species (ROS)

To characterize A375 cellular function in each treatment condition, levels of ROS were measured. A375 cells were placed in a 96-well clear-bottomed black plate (96 Well Black/Clear Bottom Plate, TC Surface, Thermo Scientific™, Waltham, MA, USA) and cultured in a medium with low glucose (5.5 mM) or high glucose (50 mM) for 24 h. Following this, each A375 cell group was treated with either dimethyl sulfoxide (DMSO) as a control, 2 mM Met, 2 mM Ime, 10 μM MF6, 10 μM ML329 or siRNA transfection for 24 h. The levels of ROS were then measured using a commercially available ROS assay kit (DOJINDO, Kumamoto, Japan). Briefly, according to the manufacturer’s instructions, cells were washed twice with Hanks’ Balanced Salt Solution (HBSS) after discarding the culture medium. Thereafter, the cells were incubated with 100 μL of Highly Sensitive DCFH-DA Working Solution for 30 min at 37 °C under normoxia conditions (95% air and 5% CO_2_). After discarding the working solution and washing twice with HBSS, fluorescence intensities were measured using a fluorescence plate reader (excitation: 490–520 nm, emission: 510–540 nm).

### 4.5. Other Analytical Methods

Total RNA extraction from the various 2D-cultured cells and subsequent reverse transcription and quantitative real-time PCR (qRT-PCR) were carried out as previously reported [45,46] using specific primers and probes (Appendix A). A two-tailed Student’s *t*-test was used to determine statistical significance for two groups, and one-way ANOVA was used to determine statistical significance for multiple groups. Significant results from one-way ANOVA were subsequently tested using Tukey’s HSD (Honestly Significant Difference) post hoc analysis. Data are shown as arithmetic means ± standard error of the mean (SEM). All statistical analyses were carried out using Graph Pad Prism 8 or 9 (GraphPad Software, San Diego, CA, USA) as described in recent reports [65,66].

## 5. Conclusions

Our findings highlight the pivotal role of redox imbalance and oxidative stress in the pathogenesis of MM. Elevated ROS levels, a hallmark of MM, were significantly reduced by Ime under high glucose conditions, whereas Met did not have this effect. Moreover, pharmacological inhibition of FABP5, FABP7 and MITF reduced ROS levels, supporting their potential as therapeutic targets. These results suggest that targeting these factors may offer novel therapeutic strategies for MM treatment. Future studies are needed to elucidate the precise mechanisms linking these factors to MM progression and drug resistance.

## Figures and Tables

**Figure 1 ijms-26-01014-f001:**
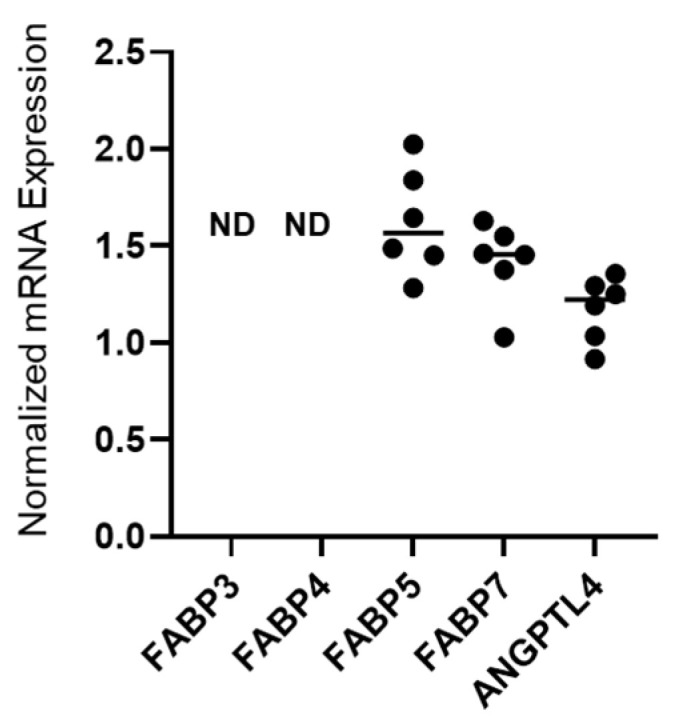
mRNA expression of *FABP* family proteins and *ANGPTL4* in A375 cells. Two-dimensionally cultured A375 cells were subjected to qPCR analysis and mRNA expression levels of *FABP3*, *FABP4*, *FABP5*, *FABP7* and *ANGPTL4* were estimated. Each gene expression was normalized with the internal control (*RPLP0* (*36B4*)). All experiments were carried out in duplicate using fresh preparations (*n* = 6 each). ND: Not detected.

**Figure 2 ijms-26-01014-f002:**
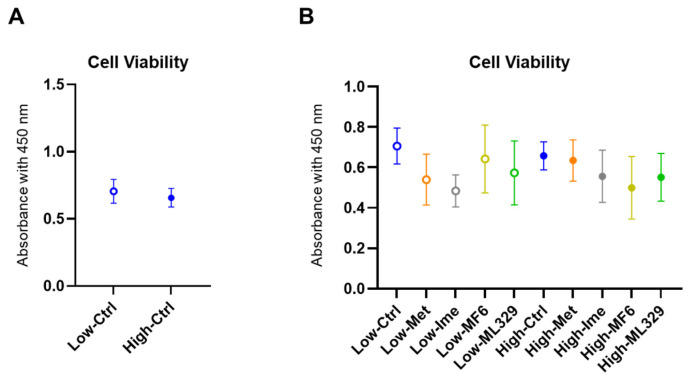
Effects of Met, Ime, MF6 and ML329 on the cell viability of A375 cells under different glucose conditions. To evaluate the cytotoxic effects of metformin (Met), imeglimin (Ime), MF6 (as a specific inhibitor of FABP5 and FABP7) and ML329 (as a specific inhibitor for MITF) on A375 cells that were cultured under low (5.5 mM) or high (50 mM) glucose conditions, living cells in 2D cultures of A375 cells were detected by using a Cell Counting Kit-8 (CCK-8) and the values plotted. (**A**): Comparison between a low glucose condition and a high glucose condition. (**B**): Comparison among agents under high and low glucose conditions. All experiments were carried out using fresh preparations (*n* = 6 each).

**Figure 3 ijms-26-01014-f003:**
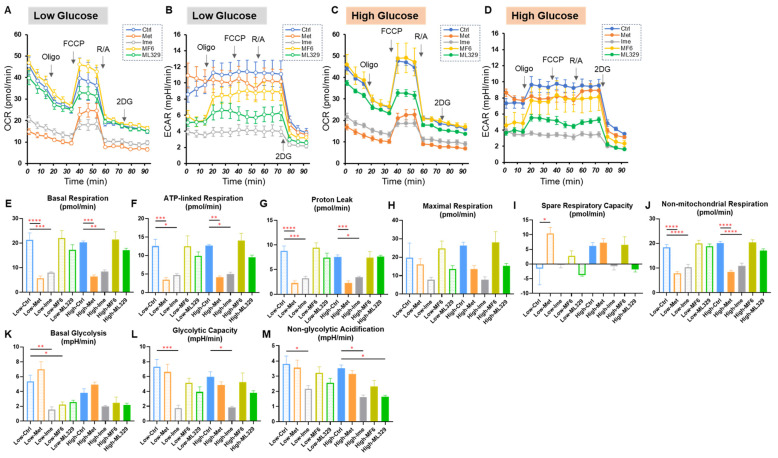
Effects of Met, Ime, MF6 or ML329 on cellular metabolic functions in A375 cells under different glucose conditions. The findings of real-time cellular metabolic function analysis of A375 cells that were treated with dimethyl sulfoxide (Ctrl), 2 mM metformin (Met), 2 mM imeglimin (Ime), 10 μM MF6 and 10 μM ML329 under low (5.5 mM) or high glucose (50 mM) conditions are shown. (**A**): plots of OCR values in A375 cells that were cultured in a low glucose condition, (**B**): plots of ECAR values in A375 cells that were cultured in a low glucose condition, (**C**): plots of OCR values in A375 cells that were cultured in a high glucose condition, (**D**): plots of ECAR values in A375 cells that were cultured in a high glucose condition, (**E**−**M**): key parameters of mitochondrial and glycolytic functions. Each parameter was calculated as below: Basal Respiration: (OCR at baseline) − (OCR with R/A), ATP-linked Respiration: (OCR at baseline) − (OCR with Oligo), Proton Leak: (OCR with Oligo) − (OCR with R/A), Maximal Respiration: (OCR with FCCP) − (OCR with R/A), Spare Respiratory Capacity: (OCR with FCCP) − (OCR at baseline), Non-mitochondrial respiration: (OCR with R/A), Basal Glycolysis: (ECAR at baseline) − (ECAR with 2DG). Glycolytic Capacity: (ECAR with Oligo) − (ECAR at baseline), Non-glycolytic Acidification: (ECAR with 2DG). OCR: oxygen consumption rate, ECAR: extracellular acidification rate, Oligo: oligomycin, FCCP: carbonyl cyanide p-trifluoromethoxyphenylhydrazone (FCCP), R/A: rotenone/antimycin A, 2-DG: 2-deoxyglucose. All experiments were performed using fresh preparations (*n* = 5−6 each). Data are presented as means ± the standard error of the mean (SEM). * *p* < 0.05, ** *p* < 0.01, *** *p* < 0.005, **** *p* < 0.001.

**Figure 4 ijms-26-01014-f004:**
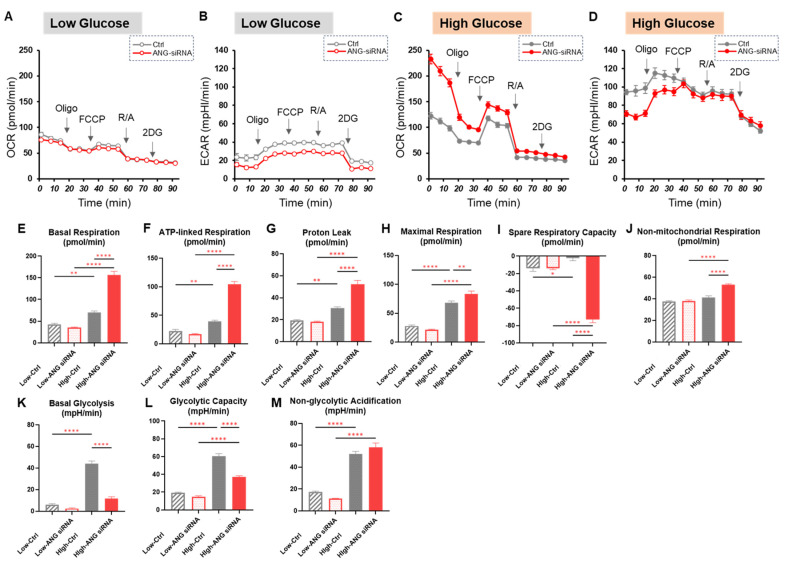
Effects of siRNA-mediated knockdown of ANGPTL4 on cellular metabolic functions in A375 cells under different glucose conditions. The results of real-time cellular metabolic function analysis of A375 cells that were transfected with ANGPTL4 siRNA (AGN-siRNA) under low (5.5 mM) and high glucose (50 mM) conditions are shown. (**A**): plots of OCR values in A375 cells that were cultured in a low glucose condition, (**B**): plots of ECAR values in A375 cells that were cultured in a low glucose condition, (**C**): plots of OCR values in A375 cells that were cultured in a high glucose condition, (**D**): plots of ECAR values in A375 cells that were cultured in a high glucose condition, (**E**−**M**): key parameters of mitochondrial and glycolytic functions. Each parameter was calculated as shown below: Basal Respiration: (OCR at baseline) − (OCR with R/A), ATP-linked Respiration: (OCR at baseline) − (OCR with Oligo), Proton Leak: (OCR with Oligo) − (OCR with R/A), Maximal Respiration: (OCR with FCCP) − (OCR with R/A), Spare Respiratory Capacity: (OCR with FCCP) − (OCR at baseline), Non-mitochondrial respiration: (OCR with R/A), Basal Glycolysis: (ECAR at baseline) − (ECAR with 2DG). Glycolytic Capacity: (ECAR with Oligo) − (ECAR at baseline), Non-glycolytic Acidification: (ECAR with 2DG). OCR: oxygen consumption rate, ECAR: extracellular acidification rate, Oligo: oligomycin, FCCP: carbonyl cyanide p-trifluoromethoxyphenylhydrazone (FCCP), R/A: rotenone/antimycin A, 2-DG: 2-deoxyglucose. All experiments were carried out using fresh preparations (*n* = 6 each). * *p* < 0.05, ** *p* < 0.01, **** *p* < 0.001.

**Figure 5 ijms-26-01014-f005:**
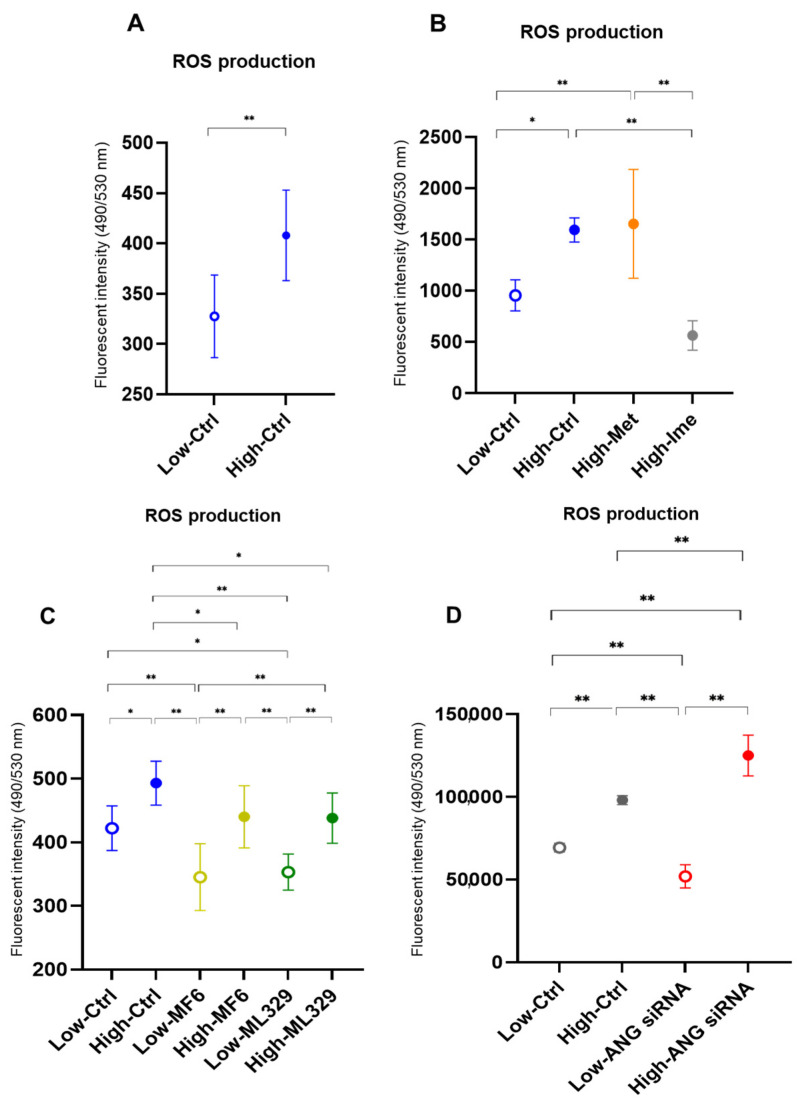
Effects of Met, Ime, MF6 and ML329 and siRNA-mediated knockdown of ANGPTL4 on levels of reactive oxygen species (ROS) in A375 cells under different glucose conditions. Planar cultured A375 cells that were treated with dimethyl sulfoxide (vehicle control), 2 mM metformin (Met), 2 mM imeglimin (Ime), 10 μM MF6 and 10 μM ML329 or transfected with control siRNA or ANGPTL4 siRNA under low (5.5 mM) and high glucose (50 mM) conditions were subjected to measurement of ROS levels and the values plotted. (**A**): Comparison between a low glucose condition and a high glucose condition. (**B**): Comparison between Met and Ime under high and low glucose conditions. (**C**): Comparison between MF6 and ML329 under high and low glucose conditions. (**D**): Comparison between control and ANGPTL4 siRNA under high and low glucose conditions. All experiments were carried out using fresh preparations (*n* = 5–6 each). * *p* < 0.05, ** *p* < 0.01.

**Figure 6 ijms-26-01014-f006:**
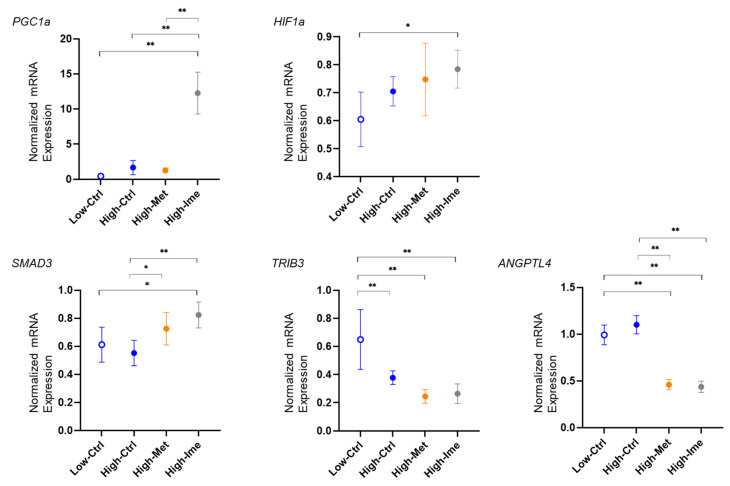
Effects of Met, Ime, MF6 and ML329 on *PGC1a*, *HIF1a*, *SMAD3* and *TRIB3* expression in A375 cells under different glucose conditions. Planar cultured A375 cells that were treated with dimethyl sulfoxide (vehicle control), 2 mM metformin (Met), 2 mM imeglimin (Ime), 10 μM MF6 and 10 μM ML329 or transfected with control siRNA or ANGPTL4 siRNA in low (5.5 mM) and high glucose (50 mM) conditions were subjected to qPCR analysis, and the relative mRNA expression levels of *PGC1a*, *HIF1a*, *SMAD3* and *TRIB3* are shown. All experiments were carried out in duplicate using fresh preparations (*n* = 6 each). * *p* < 0.05, ** *p* < 0.01.

## Data Availability

The original contributions presented in this study are included in the article/Appendix A; further inquiries can be directed to the corresponding author.

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
