# Peer review of "High-Glucose-Induced Metabolic and Redox Alterations Are Distinctly Modulated by Various Antidiabetic Agents and Interventions Against FABP5/7, MITF and ANGPTL4 in Melanoma A375 Cells"

_ijms, 2025, doi:10.3390/ijms26031014_

Round 1

Reviewer 1 Report

Comments and Suggestions for Authors

The reviewed publication "High glucose-induced metabolic and redox alterations are distinctly modulated by various antidiabetic agents and interventions against fatty acid chaperones and tumor enhancers in melanoma A375 cells." raises a very interesting topic of the relationship between different glucose levels and antidiabetic drugs on melanoma A375 cells.

In my opinion, the introduction and discussion are well written and exhaustive.

However, I have some minor comments:

- The authors should specify the goals of their experiments in more detail.

- The discussion/summary should clearly emphasize the clinical potential of the experiments, but also its limitations

- I think Figures 2, 5 and 6 would be more legible if they were prepared in many colors that would reflect the legend under the graph - these pluses and minuses are difficult to read.

- Figures 3 and 4 are very extensive. Maybe it would be worth separating them (points A-D separately from the rest), or arranging the graphics vertically (points A-D at the top, the rest underneath). I think this would significantly improve the readability of the figures.

- The Material and Methods section has been divided into subsections by the Authors. Maybe it would be worth adding subsections to the Results section?

Author Response

Dear Editor,

Thank you very much for the constructive comments concerning our manuscript “High glucose-induced metabolic and redox alterations are distinctly modulated by various antidiabetic agents and interventions against fatty acid chaperones and tumor enhancers in melanoma A375 cells.”. We carefully checked all of the Editor and reviewers’ comments and prepared a revised version of our paper that takes these comments into account. The changes are listed below.

Reviewer 1 comments

The reviewed publication "High glucose-induced metabolic and redox alterations are distinctly modulated by various antidiabetic agents and interventions against fatty acid chaperones and tumor enhancers in melanoma A375 cells." raises a very interesting topic of the relationship between different glucose levels and antidiabetic drugs on melanoma A375 cells.

In my opinion, the introduction and discussion are well written and exhaustive.

However, I have some minor comments:

  1. The authors should specify the goals of their experiments in more detail.

Answer; We sincerely appreciate your excellent comment. To specify the study aim and goal, last paragraph of Introduction and first sentence of Results are rewritten to be in alignment with paper title as follows: last paragraph of Introduction: ‘Therefore, to elucidate unidentified effects of antidiabetic regents, FABP5, ANGPTL4 and/or MITF on diabetic states of MM, high glucose-induced metabolic and redox alterations were studied using melanoma A375 cells. That is, the effects of treatment with Ime on cellular biological properties including generation of ROS and metabolic functions were compared with the effects of treatment with Met under a high glucose condition. In addition, the contribution of MITF, FABPs and ANGPTL4 to the high glucose-induced biological alterations of A375 cells was investigated.’, and first sentence of Results: ‘The purpose of the present study was to elucidate unidentified effects of antidiabetic agents, immeglimin (Ime) and metformin (Met), FABP family proteins, ANGPTL4 and MITF on biological aspects including diabetic states of MM assessed by evaluating high glucose-induced metabolic and redox alterations in melanoma A375 cells.’.

  1. The discussion/summary should clearly emphasize the clinical potential of the experiments, but also its limitations

Answer; We sincerely appreciate your excellent comment. As suggested, additional paragraph emphasizing the clinical potential of the experiments, but also its limitations in the last of Discussion: ‘We acknowledge that the current study has several limitations. Firstly, the precise linkage mechanisms by which cellular metabolic functions and levels of ROS are modulated by antidiabetic reagents, FABPs, MITF and ANGPTL4 in diabetic states of A375 cells remain to be elucidated. Secondly, we do not know why results of levels of ROS under different glucose conditions were different from the results of a previous study using MM cell lines [15]. Thirdly, the effects of antidiabetic agents, FABPs, MITF and ANGPTL4 on cellular metabolism and redox status have not characterized in detail. Fourthly, as of this writing, our results had little relevance to possibly apply to further understanding of pathogenesis and developing new therapeutic strategy for MM. Therefore, further investigation to elucidate unidentified mechanisms inducing influence of antidiabetic agents, FABPs, MITF and ANGPTL4 on diabetic states of MM by characterizing their down-stream signals using in vivo system such as diabetic animal models needs to be assessed in the future.’.

  1. I think Figures 2, 5 and 6 would be more legible if they were prepared in many colors that would reflect the legend under the graph - these pluses and minuses are difficult to read.

Answer; We sincerely appreciate your excellent comment. As pointed out, these figures are improved following excellent suggestions.

  1. Figures 3 and 4 are very extensive. Maybe it would be worth separating them (points A-D separately from the rest), or arranging the graphics vertically (points A-D at the top, the rest underneath). I think this would significantly improve the readability of the figures.

Answer; We sincerely appreciate your excellent comment. As pointed out, these figures are improved following excellent suggestions.

  1. The Material and Methods section has been divided into subsections by the Authors. Maybe it would be worth adding subsections to the Results section?

Answer; We sincerely appreciate your excellent comment. As suggested, Results are divided into subsections.

Reviewer 2 Report

Comments and Suggestions for Authors

In presented study researchers tested the effect of Imeglimin аnd Metformin on malignant melanoma cell line A375 in condition of normal and high glucose levels. Also, impact on ROS production was tested and related gene expression. I have few major comments:

-It is unusual that title contain words chaperons and enhancers and after that these terms were not mentioned in Introduction or Discussion. 

-Discussion should be extended, it seems like it's ending abruptly.

-Please re-write section 4.3 because it is almost completely copy/paste from your previous paper.

-The aims defined in last paragraph of Introduction should be in alignment with paper title and first sentence of Results.

Minor comments:

-Fig3 will be more readible if you split presented graphics in two rows (e.g. one row A, B, C, and D, and below second row E, F, G, H, I, J, K, L, M, and N)

-Ln203-206 and how you explain that? That is difference from previous study on MM?

-Ln221 ROS already introduced

-Ln246 typo Met, Ime

-Ln271 introduce abbreviation T2DM

-Ln319 uniform abbreviation for USA (U.S.A)

-Ln307-309 why you apply for IRB approval for exp with cell line (especially commercial available cell line)? And how Declaration of Helsinki is correlated with exp on cell line? Further in Ln387 that IRB Statement is Not applicable

Author Response

Dear Editor,

Thank you very much for the constructive comments concerning our manuscript “High glucose-induced metabolic and redox alterations are distinctly modulated by various antidiabetic agents and interventions against fatty acid chaperones and tumor enhancers in melanoma A375 cells.”. We carefully checked all of the Editor and reviewers’ comments and prepared a revised version of our paper that takes these comments into account. The changes are listed below.

Reviewer 2 comments

In presented study researchers tested the effect of Imeglimin аnd Metformin on malignant melanoma cell line A375 in condition of normal and high glucose levels. Also, impact on ROS production was tested and related gene expression. I have few major comments:

  1. It is unusual that title contain words chaperons and enhancers and after that these terms were not mentioned in Introduction or Discussion.

Answer; We sincerely appreciate your excellent comment. As pointed out, words of chaperons and enhancers in title was not suitable, and therefore, title is changed to’ High glucose-induced metabolic and redox alterations are distinctly modulated by various antidiabetic agents and interventions against FABP5/7, MITF and ANGPTL4 in melanoma A375 cells.’

  1. Discussion should be extended, it seems like it's ending abruptly.

Answer; We sincerely appreciate your excellent comment. As suggested, additional paragraph emphasizing the clinical potential of the experiments, but also its limitations in the last of Discussion: ‘We acknowledge that the current study has several limitations. Firstly, the precise linkage mechanisms by which cellular metabolic functions and levels of ROS are modulated by antidiabetic reagents, FABPs, MITF and ANGPTL4 in diabetic states of A375 cells remain to be elucidated. Secondly, we do not know why results of levels of ROS under different glucose conditions were different from the results of a previous study using MM cell lines [15]. Thirdly, the effects of antidiabetic agents, FABPs, MITF and ANGPTL4 on cellular metabolism and redox status have not characterized in detail. Fourthly, as of this writing, our results had little relevance to possibly apply to further understanding of pathogenesis and developing new therapeutic strategy for MM. Therefore, further investigation to elucidate unidentified mechanisms inducing influence of antidiabetic agents, FABPs, MITF and ANGPTL4 on diabetic states of MM by characterizing their down-stream signals using in vivo system such as diabetic animal models needs to be assessed in the future.’.

  1. Please re-write section 4.3 because it is almost completely copy/paste from your previous paper.

Answer; We sincerely appreciate your excellent comment. As pointed out, corresponding method is revised to avoid similarity with our previous paper.

  1. The aims defined in last paragraph of Introduction should be in alignment with paper title and first sentence of Results.

Answer; We sincerely appreciate your excellent comment. As suggested, last paragraph of Introduction and first sentence of Results are rewritten to be in alignment with paper title as follows: last paragraph of Introduction: ‘Therefore, to elucidate unidentified effects of antidiabetic regents, FABP5, ANGPTL4 and/or MITF on diabetic states of MM, high glucose-induced metabolic and redox alterations were studied using melanoma A375 cells. That is, the effects of treatment with Ime on cellular biological properties including generation of ROS and metabolic functions were compared with the effects of treatment with Met under a high glucose condition. In addition, the contribution of MITF, FABPs and ANGPTL4 to the high glucose-induced biological alterations of A375 cells was investigated.’, and first sentence of Results: ‘The purpose of the present study was to elucidate unidentified effects of antidiabetic agents, immeglimin (Ime) and metformin (Met), FABP family proteins, ANGPTL4 and MITF on biological aspects including diabetic states of MM assessed by evaluating high glucose-induced metabolic and redox alterations in melanoma A375 cells.’.

Minor comments:

  1. Fig3 will be more readible if you split presented graphics in two rows (e.g. one row A, B, C, and D, and below second row E, F, G, H, I, J, K, L, M, and N)

Answer; We sincerely appreciate your excellent comment. As pointed out, these figures are improved following excellent suggestions.

  1. Ln203-206 and how you explain that? That is difference from previous study on MM?

Answer; We sincerely appreciate your excellent comment. In terms of the difference of levels of ROS under different glucose conditions between our present study and previous study, although I honestly do not know why, the difference may be related to different exposure period (ours: 24 hours vs previous: 48 hours) to different glucose concentration (ours: 5 mM and 50 mM vs previous: 5 mM and 25 mM) and using different detection kit for ROS. Therefore, this information is included in the study limitation in the Discussion; ‘We acknowledge that the current study has several limitations. Firstly, the precise linkage mechanisms by which cellular metabolic functions and levels of ROS are modulated by antidiabetic reagents, FABPs, MITF and ANGPTL4 in diabetic states of A375 cells remain to be elucidated. Secondly, we do not know why results of levels of ROS under different glucose conditions were different from the results of a previous study using MM cell lines [15]. Thirdly, the effects of antidiabetic agents, FABPs, MITF and ANGPTL4 on cellular metabolism and redox status have not characterized in detail. Fourthly, as of this writing, our results had little relevance to possibly apply to further understanding of pathogenesis and developing new therapeutic strategy for MM. Therefore, further investigation to elucidate unidentified mechanisms inducing influence of antidiabetic agents, FABPs, MITF and ANGPTL4 on diabetic states of MM by characterizing their down-stream signals using in vivo system such as diabetic animal models needs to be assessed in the future.’.  

  1. Ln221 ROS already introduce

Answer; We sincerely appreciate your excellent comment. As pointed out, repeated full spelling is omitted.

  1. Ln246 typo Met, Ime

Answer; We sincerely appreciate your excellent comment. As pointed out, this type error is corrected.

  1. Ln271 introduce abbreviation T2DM

Answer; We sincerely appreciate your excellent comment. As pointed out, this is changed to ‘type 2 diabetes (T2DM)’.

  1. Ln319 uniform abbreviation for USA (U.S.A)

Answer; We sincerely appreciate your excellent comment. As pointed out, this is corrected.

  1. Ln307-309 why you apply for IRB approval for exp with cell line (especially commercial available cell line)? And how Declaration of Helsinki is correlated with exp on cell line? Further in Ln387 that IRB Statement is Not applicable

Answer; We sincerely appreciate your excellent comment. We had permission of IRB approval in which our experiments using various sources of cells were approved even if we used commercially available cell lines. However, as pointed out, Declaration of Helsinki is not needed because we used only commercially available cell line and therefore, this phrase is deleted.

Round 2

Reviewer 2 Report

Comments and Suggestions for Authors

Dear Authors,

I am satisfied with manuscript improvement.